# Effects of Plasma-Activated Water Treatment on the Inactivation of Microorganisms Present on Cherry Tomatoes and in Used Wash Solution

**DOI:** 10.3390/foods12132461

**Published:** 2023-06-23

**Authors:** Gaeul Lee, Sung-Wook Choi, Miyoung Yoo, Hyun-Joo Chang, Nari Lee

**Affiliations:** 1Food Safety and Distribution Research Group, Korea Food Research Institute, 245, Nongsaengmyeong-ro, Wanju-gun 55365, Jeollabuk-do, Republic of Korea; 50045@kfri.re.kr (G.L.); swchoi@kfri.re.kr (S.-W.C.); hjchang@kfri.re.kr (H.-J.C.); 2Food Standard Research Group, Korea Food Research Institute, 245, Nongsaengmyeong-ro, Wanju-gun 55365, Jeollabuk-do, Republic of Korea; myyoo@kfri.re.kr

**Keywords:** plasma-activated water, microbial inactivation, cherry tomato, wash solution

## Abstract

Herein, we investigated the potential of plasma-activated water (PAW) as a wash solution for the microbial decontamination of cherry tomatoes. We analyzed the efficacy of PAW as a bactericidal agent based on reactive species and pH. Immersion for 5 min in PAW15 (generated via plasma activation for 15 min) was determined as optimal for microbial decontamination of fresh produce. The decontamination efficacy of PAW15 exceeded those of mimic solutions with equivalent reactive species concentrations and pH (3.0 vs. 1.7 log reduction), suggesting that the entire range of plasma-derived reactive species participates in decontamination rather than a few reactive species. PAW15-washing treatment achieved reductions of 6.89 ± 0.36, 7.49 ± 0.40, and 5.60 ± 0.05 log_10_ CFU/g in the counts of *Bacillus cereus*, *Salmonella* sp., and *Escherichia coli* O157:H7, respectively, inoculated on the surface of cherry tomatoes, with none of these strains detected in the wash solution. During 6 days of 25 °C storage post-washing, the counts of aerobic bacteria, yeasts, and molds were below the detection limit. However, PAW15 did not significantly affect the viability of RAW264.7 cells. These results demonstrate that PAW effectively inactivates microbes and foodborne pathogens on the surface of cherry tomatoes and in the wash solution. Thus, PAW could be used as an alternative wash solution in the fresh produce industry without cross-contamination during washing and environmental contamination by foodborne pathogens or potential risks to human health.

## 1. Introduction

The market of fresh produce and minimally processed produce has resulted in a high demand from consumers looking for fresh, healthy, convenient, and ready-to-eat foods [1,2]. Cherry tomatoes, a tomato variety, are preferred by fresh produce consumers because, like other tomato varieties, they provide abundant nutrition, have a delicious taste, and are relatively more manageable to serve due to their small size. However, cherry tomato (*Solanum lycopersicum* var. *cerasiforme*), which is generally consumed uncooked, is highly likely to be infected with foodborne pathogens, which can contaminate fresh produce via many routes such as irrigation water, organic fertilizer, and harvest and handling conditions [3]. *Escherichia coli*, *Salmonella* sp., *Listeria monocytogenes*, and *Bacillus cereus* have been implicated in foodborne outbreaks associated with the consumption of fresh produce, e.g., leafy vegetables, tomatoes, apples, melon, spinach, lettuce, sprouts, and fresh cut salad [3,4,5,6]. Commonly encountered spoilage microorganisms such as *Pseudomonas* sp., coryneform bacteria, Enterobacteriaceae, lactic acid bacteria, mold, and yeasts can contaminate fresh produce [7,8]. Thus, ensuring the safety of fresh produce from microbial contamination continues to be a primary concern in many countries.

Sanitizer wash is the primary step practiced in the fresh produce industry to reduce field-acquired dirt and microbial contamination and improve quality and shelf life. In the fresh produce industry, the produce undergoes three washing steps. The first and second steps involve washing with tap water to remove dirt and with a sanitizer solution, respectively. The final washing step uses tap water to reduce sanitizer odor [9]. Chlorine-based sanitizers are the most commonly used to lessen the risk of foodborne pathogen contamination, being relatively low-cost, effective, and straightforward to use [10]. However, several concerns remain regarding human health when using chlorinated organic compounds, such as unpleasant odors in fresh produce and environmental pollution that have led to a search for alternatives [11,12]. Plasma-activated water (PAW), recently reported to inactivate a variety of microorganisms [13,14], could be an alternative to chlorine-based sanitizers in the industrial washing of fresh produce, eliminating the need for the third additional rinsing step to remove sanitizers.

The microbial inactivation effects of PAW are a result of reactive nitrogen species (RNS), reactive oxygen species (ROS), and other active particles (electrons and ions), mainly generated during plasma discharge in water [15]. The dissolution of plasma-generated substances in water leads to the formation of short-lived transient reactive species, e.g., hydroxyl (OH) and nitrogen dioxide (NO_2_) radicals, and long-lived stable reactive species, e.g., hydrogen peroxide (H_2_O_2_), nitrite (NO_2_^−^), and nitrate (NO_3_^−^) ions. While ROS, such as H_2_O_2_, cause oxidizing reactions in microbial cells, thereby disrupting the cell membrane [16], RNS, such as nitric oxide and its derivatives, including NO_2_^−^, NO_3_^−^, and peroxynitrite, contribute to acidification, thereby inducing cell death [14,17].

Recent reports have shown that PAW could inactivate foodborne pathogens inoculated on fresh produce, such as strawberries, grape berries, tomatoes, and mushrooms, and maintain their quality [15,18,19,20]. Studies have reported that PAW can also effectively inactivate microbes commonly present on the surface of fresh produce in nature, including aerobic bacteria, yeasts, and molds, helping to increase the shelf-life of fresh produce and ensuring safety and quality during storage [21,22,23,24]. However, there are limited reports on the applicability of PAW as a wash solution in the fresh produce industry and much less information on whether the application of PAW, effective against a broad range of pathogens, is safe for human health.

Through this study, we aimed to verify the feasibility of applying PAW in the washing process in the fresh produce industry, and cherry tomatoes were used as the target fresh produce. We analyzed the concentrations of reactive species present in PAW, and based on this information, we evaluated the microbial inactivation efficacy of PAW and artificial solutions mimicking the concentrations of reactive species in PAW. We assessed the effectiveness of the PAW-washing treatment against foodborne pathogens inoculated on the surface of cherry tomatoes or present in used wash solutions and investigated changes in microbial counts on the surface of PAW-washed cherry tomatoes during storage. In addition, we examined the cytotoxicity of PAW using a mammalian cell line.

## 2. Materials and Methods

### 2.1. Bacterial Strains

Three bacterial strains, i.e., *B. cereus* ATCC 14579, *Salmonella enterica* subsp. *enterica* serovar Typhimurium ATCC 43971, and *E. coli* O157:H7 NCCP 1109, were used in this study. *B. cereus* especially is not a common pathogen on tomatoes and few cases have been associated with food illness outbreaks in tomatoes. However, *B. cereus* is widely distributed in environments such as soil and plants and is frequently isolated from tomatoes, even though the contamination level was less than 1 log CFU/g. Therefore, it was used as target bacteria in this study as potential risk bacteria. A single colony of each strain was grown overnight in Luria–Bertani (Difco™, Sparks, MD, USA) or tryptic soy broth (Difco™) with shaking at 37 °C, and the cultures were used for inoculating cherry tomatoes.

### 2.2. Fresh Produce

Cherry tomatoes (*Solanum lycopersicum* var. *cerasiforme*), a widely consumed fresh produce, were purchased from a smart farm in Wanju, Jeollabuk-do, harvested at the red maturity stage, and stored in a refrigerator (4 °C). Among them, the cherry tomatoes weighing approximately 17 ± 1 g, bearing intact stems, were used in further experiments.

### 2.3. Preparation of PAW

The equipment used for generating PAW in this study is shown in Figure 1. After injecting 120 mL of distilled water (DW) into an acrylic plasma chamber (200 mL), PAW was generated using a bubble plasma jet (SD-A/G-1; Plasma Holdings Co., Ltd., Changwon, Republic of Korea). For discharge in the plasma chamber, a mixed gas (70% nitrogen and 30% oxygen) was injected along the cathode electrode at a flow rate of 10–12 L/min in a quartz capillary tube (internal diameter 5 mm) to form a plasma using 4~7 kV DC (30 mW). For supplying this high voltage, a power supply unit was designed to use 110 V AC (50~60 Hz) as input voltage of 110 V AC (50~60 Hz), capable of providing an output of 3~30 KV (0~10 mA). To treated the cherry tomatoes with 2000 mL PAW, we used 10 PAW devices, each producing 120 mL of PAW, in parallel.

### 2.4. Quantification of pH Values and the Dissolved Concentrations of H_2_O_2_, NO_2_^−^, and NO_3_^−^ in PAW

The physicochemical characteristics of PAW at different activation times (5, 10, 15, 20, and 30 min) were analyzed immediately after PAW generation. The pH values were measured using a pH meter (Mettle Toledo, Columbus, OH, USA) connected to a pH probe. The concentration of H_2_O_2_ in PAW was measured using the titanium oxysulfate assay (TiOSO_4_; Sigma-Aldrich, St. Louis, MO, USA) employing a spectrophotometer (Biospectrometer; Eppendorf, Hamburg, Germany). Samples were prepared for the titanium oxysulfate assay by adding 1 mL of sample, 0.1 mL of azide solution (60 mM NaN_3_), and 0.5 mL of titanium oxysulfate–sulfuric acid solution. The absorbance was measured at 409 nm [25]. The concentration of NO_2_^−^ was measured using the colorimetric assay employing Griess reagent (Sigma-Aldrich, 1465-25-4). Approximately 80 μL of buffer solution (20 mM, pH 7.6) was added directly to 20 μL of the sample in a 96-well plate. Then, 50 μL of Griess Reagent A was added, and the culture plate was incubated at 25 °C. After 5 min, 50 μL of Griess Reagent B was added, and, following a 10 min incubation at 25 °C, the absorbance was measured at 540 nm using a microplate reader (SpectraMaxi3X; Molecular Devices Corp., San Jose, CA, USA) [26]. The concentration of NO_3_^−^ was measured using a nitrate assay kit (Spectroquant^®^ test kits; Merck, Darmstadt, Germany). Following the manufacturer’s instructions, 0.4 mL of PAW was sampled and mixed with reagents NO_3_-1 and NO_3_-2. After 10 min of reaction, NO_3_^−^ concentrations were determined by measuring the absorbance at 540 nm in the microplate reader (SpectraMaxi3X). The change of pH, oxidation-reduction potential and electrical conductivity were measured using a multi-meter (HI 98194, Hanna Instruments, Woonsocket, RI, USA) and are shown in Appendix A.

### 2.5. Optimal Conditions for PAW Treatment

Approximately 50 μL of *B. cereus* cultures were added to 4.95 mL of freshly generated PAW immediately after 15 min of plasma activation to determine the optimal contact time (holding time, HT) between PAW and the microorganisms. After 5, 10, 15, 20, and 30 min of HT, 0.1 mL of each of these cultures exposed to different HTs or their diluted aliquots were spread-plated onto plate count agar (PCA; Difco™) medium for microbial counting. To evaluate the optimal plasma activation time for generating PAW, 4.95 mL of water exposed to plasma for 5, 10, 15, 20, and 30 min were added to 50 μL aliquots of *B. cereus* cultures. After 5 min of HT, 0.1 mL of the PAW-treated cultures were used for microbial counting. Sterile DW-treated *B. cereus* cultures were used as controls to determine initial bacterial concentrations. All agar media were incubated at 37 °C for 18–24 h before colonies were counted. The microbial counts are expressed as log_10_ CFU/mL of the sample.

### 2.6. Microbial Inactivation Effects of Artificially Generated Reactive Species

Sodium nitrite (Sigma-Aldrich), sodium nitrate (Sigma-Aldrich), and H_2_O_2_ solution (Sigma-Aldrich) were dissolved in DW to achieve the desired concentrations, and pH was adjusted using sodium nitrate. These mimic solutions and their mixes were added to a total volume of 5 mL in 50 μL *B. cereus* cultures. After 5 min of HT, 0.1 mL of cultures treated with these artificial solutions were used for microbial counting. Cell counts were determined via serial dilutions of suspensions and subsequent enumeration on PCA. The microbial counts are expressed as log_10_ CFU/mL of the sample.

### 2.7. PAW-Induced Inactivation of Foodborne Pathogens Inoculated on the Surface of Cherry Tomatoes

Cherry tomatoes were washed with 70% ethanol, rinsed with sterile DW, and irradiated with ultraviolet light for 2 h to remove contaminants. Approximately 0.02 mL of each bacterial inoculum was spread on the surface of cherry tomatoes and dried in a biosafety hood for 1 h 30 min. Cherry tomatoes were placed into a sterile stomacher bag without filter (Whirl-Pak^®^; Nasco Co., Fort Atkinson, WI, USA) containing PAW15 or sterile DW, sealed with an Impulse bag sealer (Lovero SK-210; Sambo Tech Co., Ltd., Gimpo, Republic of Korea) and washed by immersion for 5 min. Sterile DW was used as the wash solution to prevent contamination from external sources. Cherry tomatoes treated using PAW or sterile DW (control) were placed into a sterile stomacher bag with a filter (Whirl-Pak^®^) containing 5 mL Dulbecco’s phosphate-buffered saline (DPBS; Sigma-Aldrich), and the samples were homogenized manually. After homogenization, 1 mL aliquots of samples were serially diluted tenfold in 9 mL of sterile DPBS, and 1 mL of the homogenized samples or their diluted aliquots were spread onto each selective medium. Likewise, 1 mL of wash solution or diluted aliquots were spread as described above. Microfast^®^ *B. cereus* Count Plate (Meizheng Bio-Tech, Rizhao, China), SS MC-Media Pad™ (JNC Corp., Tokyo, Japan), and Microfast^®^ Coliform & *E. coli* Count Plate (Meizheng Bio-Tech) were used as selective media for estimating the cell counts of *B. cereus*, *S*. Typhimurium., and *E. coli* O157:H7, respectively. The count plates were incubated at 37 °C for 24 h, and viable cells were counted. All procedures were performed in a biosafety hood. The microbial counts are expressed as log_10_ CFU/g of the sample.

### 2.8. Changes in the Counts of Native Microbes Present on the Surface of PAW-Washed Cherry Tomatoes during Storage

The cherry tomatoes were divided randomly into three groups, of which one group was immersed for 5 min in a 3000 mL sterile beaker containing 2000 mL of PAW15, another group was immersed in 2000 mL sterile DW, and the third group was used as a control without any treatments. The treated fruits were placed individually on sterile polyethylene food wrap films and air-dried inside a biosafety hood for 30 min at room temperature. Subsequently, all samples were stored for 6 days in an incubator at 25 °C, under 90% relative humidity. Cherry tomatoes from each group were randomly selected for microbiological analysis on day 0 and at 2-day intervals during storage. For microbial cell count analysis, three cherry tomatoes from each treatment group were transferred aseptically into sterile stomacher bags (Whirl-Pak^®^) with 35 mL of sterile DPBS (Sigma-Aldrich) and homogenized manually by rubbing for 5 min. Approximately 1 mL aliquots of the homogenate taken from the stomacher bags were serially diluted tenfold in sterile DPBS (Sigma-Aldrich). The dilutions were spread onto Petrifilm aerobic count plates (3M™, St. Paul, MN, USA) plates and Petrifilm yeast/ mold count plates (3M™) and incubated at 35 and 25 °C for the subsequent counting of CFU of bacteria and fungi, respectively. The microbial counts are expressed as log_10_ CFU/g of the sample. 

### 2.9. Cytotoxicity Test

Cell viability was measured via a 3-(4,5-dimethylthiazol-2-yl)-2,5-diphenyltetrazolium bromide (MTT) assay [27]. Briefly, RAW264.7 cells were seeded in a 96-well plate at a cell density of 2 × 10^4^ cells/ well and incubated for 24 h at 37 °C in a 5% CO_2_ incubator. Cells were treated with 10.0 μL of plasma-treated water. The MTT solution (5 mg/mL) was added to each well after 24 h of incubation, and the plates were incubated for another 4 h at 37 °C. Approximately 0.1 mL of dimethyl sulfoxide was added to dissolve formazan post-removal of the cell supernatant. The absorbance was measured at 540 nm using a microplate reader (SpectraMaxi3x). Cell viability is expressed as a percentage of the absorbance of the treated cells compared with that of the untreated cells.

### 2.10. Statistical Analyses

All experiments were replicated three or five times. Values from all experiments are expressed as mean ± standard deviation. Data were analyzed using Prism software v4.03 for Windows (GraphPad, San Diego, CA, USA) as needed.

## 3. Results and Discussion

### 3.1. Optimal Characteristics of PAW Required for Efficient Microbial Inactivation

Plasma-generated reactive species are a crucial factor affecting PAW-induced microbial inactivation, as demonstrated previously [25,26]. Therefore, it is necessary to determine the concentrations and types of reactive species in PAW, varying due to factors including plasma source, plasma type, and gases and solutions used in the treatment, and to optimize the device and conditions used to generate suitable reactive species [28].

After DW was treated with plasma for different times (5, 10, 15, 20, and 30 min), the concentrations of H_2_O_2_, NO_2_^−^, and NO_3_^−^ in the generated PAW were quantitatively measured and pH, the measure of the hydrogen ion (H^+^) concentration of a solution, was also recorded. As shown in Figure 2, the pH of PAW rapidly decreased to pH 3.39 ± 0.21 during the first 5 min and reached a steady state of 3.01 ± 0.25 after 15 min of plasma treatment. The concentrations of H_2_O_2_ and NO_3_^−^ in PAW gradually increased with increasing lengths of treatment times. The species concentration of NO_2_^−^ rapidly increased to 0.198 ± 0.0070 mM after 5 min of plasma treatment and reached 0.228 ± 0.0062 mM after 10 min, but rapidly decreased after the 15 min treatment. The reduction in NO_2_^−^ concentrations after the 15 min treatment can be ascribed to the peroxynitration process that occurs in DW with a pH lower than 4 after plasma treatment. Several researchers have reported that reactive species such as H_2_O_2_, nitric acid (HNO_3_), and peroxynitrous acid are formed during the PAW generation process, resulting in low pH and antibacterial activity in PAW [13,14,29,30].

In this process, oxidation-reduction potential (ORP) levels, which represent chemical strength, reached 545 mV at 5 min after plasma discharge, increased to 630 mV after 20 min, and maintained 640~660 mV until 60 min. Similar results have also been reported for the PAW formed by air plasma jet [19,31] and air DBD plasma [32]. These results mean that oxidizing species are being produced in PAW, and species that are more powerful than in the PAW generated using argon or oxygen as a flow gas peroxynitrous acid [19]. Electrical conductivity (EC) is often used as an important indicator of the concentration of ions existing in water. It increased gradually during plasma discharge, increasing from 12 to 980 μS/cm at 20 min after discharge, and reached 3.8 mS/cm at 60 min after discharge. This indicated that various reactive species were generated in water depending on the plasma treatment time. The results of ORP and EC are shown in Appendix A.

Our results indicate that PAW generated via 15 min plasma treatment is suitable for investigating the antimicrobial effects of PAW because it has sufficient concentrations of reactive species and a low pH value to induce this effect.

The *B. cereus* strain was exposed to PAW for varying time lengths to evaluate the effects of varying HTs on bacterial counts. As shown in Figure 3a, treatment with PAW15 (generated via plasma activation for 15 min) caused at least a 3-log reduction in the cell counts of *B. cereus* after 5 min of HT, whereas the counts of viable *B. cereus* cells decreased to 1.11 ± 0.12 log_10_ CFU/mL after 10 min of HT. These results are consistent with those of previous studies [15,22,33,34], i.e., the longer the contact time, i.e., HT between bacteria and PAW, the higher the bactericidal effect. However, it is necessary to determine the optimal HT for achieving maximum inactivation efficiency with minimum exposure to PAW, as the reactive species present in PAW can reduce the quality of the fresh produce. Many studies recommend the duration of treatment to be restricted to <5 min for minimal variation in quality [21,22]. The Ministry of Food and Drug Safety in Korea also suggests immersion for 5 min, based on the sodium hypochlorite treatment duration, as a guideline for the use of disinfectants in food. Therefore, it is reasonable to restrict the washing duration to 5 min in this study because 5 min of HT reduced the bacterial counts by approximately 3-log CFU without affecting the quality of the fresh produce. 

Based on our results that the concentrations of reactive species present in PAW depend on the duration of the plasma activation time for PAW generation, we analyzed whether the antibacterial efficiency of PAW is influenced by the plasma treatment duration (Figure 3b). After PAW15-treatment, approximately a 3-log reduction was achieved in the count of *B. cereus* after 5 min of HT, whereas treatment with PAW30 (generated via 30 min of plasma treatment) caused complete inactivation of *B. cereus*. We expected that peroxynitrous acid, which significantly affects antibacterial activity, would increase depending on the species concentration of NO_2_^−^ [34,35,36], such that PAW10 or PAW15 would possess the highest bactericidal effect. However, the results showed that the microbial inactivation efficiency of PAW increased as the plasma treatment time for PAW generation increased.

### 3.2. Microbial Inactivation Based on the Composition of Artificial Reactive Species

Several studies [29,37,38] have reported that PAW exhibits bactericidal effects mainly caused by reactive species, such as H_2_O_2_, NO_2_^−^, and NO_3_^−^, and low pH values. The quantitative detections of H_2_O_2_, NO_2_^−^, and NO_3_^−^ could represent a primary indicator for plasma source characterization regarding the antimicrobial activity associated with PAW [39]. Tarabová et al. [40] reported that H_2_O_2_ and NO_2_^−^ generated by plasma discharge in water produce peroxynitrous acid under acidic conditions. Approximately 70% of peroxynitrous acid is converted to HNO_3_ to produce NO_3_^−^ and H^+^, whereas the remaining 30% is broken down into OH and NO_2_ radicals, causing microbial inactivation. 

Our result in Section 3.1 indicates the possibility of additional factors affecting antimicrobial activity other than the well-known reactive species or the products produced in PAW. To determine if other factors exist, we measured the antimicrobial activity against *B. cereus* in mimic solutions prepared using various combinations of reactive species comparable to their concentrations in PAW15 (Figure 4c). As a preliminary experiment, the effects of varying pH values and H_2_O_2_ concentrations on the bactericidal activity of mimic solutions were analyzed. The results showed that the viable cell counts of *B. cereus* remained unchanged at a 2 mM concentration of H_2_O_2_ and at concentrations equivalent to that in PAW15 (Figure 4a). However, treatment with mimic solutions reduced the cell counts of *B. cereus* by 0.90 log_10_ CFU/mL, but only by lowering the pH to approximately 2.8 (Figure 4b).

Furthermore, in a separate experiment, no reduction was observed in the counts of *B. cereus* treated separately or using a mix of nitrous acid (HNO_2_) and HNO_3_ solutions at HNO_2_ and HNO_3_ concentrations equivalent to those generated by 15 min of plasma treatment (Figure 4c). For solutions containing the combination of artificial reactive species, the decrease in the viable cell counts of *B. cereus* appeared only at a pH adjusted to 3.0, with a reduction of 1.70 log_10_ CFU/mL observed in the solution mix containing all reactive species identified in PAW15 (Figure 4c). Nevertheless, the antimicrobial effect of this mimic solution was lower than that of PAW15 (3 log reduction). This difference in the antimicrobial effects of the two solutions could be attributed to the synergistic effect caused by peroxynitrous acid and other factors inherent in PAW. We indirectly evaluated the formation of peroxynitrous acid as a critical bactericidal agent, with the concentrations of peroxynitrite precursors H_2_O_2_ and NO_2_^−^. Otherwise, peroxynitrous acid may have disappeared during the preparation of the mimic solutions; it could not be ascertained whether the reduction in NO_2_^−^ concentration (Figure 2b) necessarily led to a reduction in the formation of peroxynitrous acid. This result illustrates that the total bactericidal effect associated with PAW may result from the synergistic activity of diverse short- and long-lived species and can also be related to the diverse range of charged particles resulting from a plasma discharge [34,38,41]. Thus, the PAW generated in this study had a higher sterilization capacity dependent on the plasma discharge time due to the generation of long-lived H_2_O_2_, NO_2_^−^, and NO_3_^−^ species and other particles, as well as short-lived OH and NO_2_ radicals produced by the decomposition of peroxynitrous acid.

### 3.3. Inactivation of Foodborne Pathogens Present on the Surface of Cherry Tomatoes and in the Wash Solution

To evaluate whether PAW could inactivate foodborne pathogens present on the surface of cherry tomatoes, we tested the changes in the counts of bacterial populations present on cherry tomatoes and within the used wash solution post-washing, and the results are shown in Figure 5. After washing by immersion for 5 min in PAW15, *B. cereus*, *Salmonella* sp., and *E. coli* O157:H7 inoculated on the surface of cherry tomatoes showed a reduction of 6.89 ± 0.36, 7.49 ± 0.40, and 5.60 ± 0.05 log_10_ CFU/g in their viable cell counts, respectively (Figure 5a). Compared with *B. cereus*, *Salmonella* sp., and *E. coli* O157:H7 inoculated cherry tomatoes treated with sterile DW for the same length of time, PAW15 treatment achieved a significantly higher reduction of approximately 1.74 log_10_ CFU/g (*p* ≤ 0.05), 2.66 log_10_ CFU/g (*p* ≤ 0.005), and 1.64 log_10_ CFU/g (*p* ≤ 0.05), respectively (Figure 5a).

This result indicates that PAW can be applied as a wash solution in the fresh produce industry. However, it is necessary to increase the microbial inactivation efficacy of PAW and shorten the wash duration for ensuring practicality in industrial applicability. Joshi et al. [13] reported that PAW treatment for 3 min with agitation showed a microbial inactivation effect comparable to that obtained with the treatment for 10 min without agitation. A few studies have demonstrated that the application of micro or air bubbles in the food industry may improve the cleansing efficacy of washing processes by positively affecting the detachment and inactivation of bacteria in fresh produce [42,43,44]. We evaluated the inactivation effects of PAW treatment alone on foodborne pathogens present on the surface of cherry tomatoes without any external physical factors, suggesting that the inactivation efficacy of PAW treatment for foodborne pathogens could be potentially improved when assisted by microbubbles or agitation in the washing process of cherry tomatoes. After sterile DW treatment, viable cell counts of *Salmonella* sp., *E. coli* O157:H7, and *B. cereus* were 2.78 ± 0.24, 5.20 ± 0.07, and 3.79 ± 0.01 log_10_ CFU/g, respectively, in the used wash solution (Figure 5b). After PAW treatment, none of these bacterial strains could be detected in the used wash solutions, as PAW treatment induced a complete inactivation of these microbes in the used wash solutions (Figure 5b). In the fresh produce industry, no bacteria remaining in the used wash solutions can be a great advantage, preventing recontamination of fruits and vegetables by foodborne pathogens that detach from fruits or vegetables during the washing process while reducing the risk of environmental contamination with these pathogens. Therefore, PAW can play a major role in the washing process of fresh produce as it does not require additional chemical reagents, thereby reducing costs by omitting the third rinsing step for removing the disinfectants.

### 3.4. Inactivation of Native Microbes Present on the Surface of Cherry Tomatoes

Considering the room temperature distribution of fresh produce locally, changes in viable cell counts of native microbiota present on the surface of cherry tomatoes washed with PAW were analyzed during storage (6 days) at room temperature. A comparative analysis of the microbial counts in cherry tomatoes in the PAW-treated group, the unwashed group, and the group washed with sterile DW is presented in Figure 6. Initial counts of total aerobic bacteria in the unwashed control group and sterile DW-washed group were 1.57 ± 0.63 and 1.17 ± 0.14 log_10_ CFU/g, respectively. An insignificant increase of <0.6 log_10_ CFU/g was observed in the final counts of aerobic bacteria in both groups as compared to the initial counts, likely due to the low levels of native bacteria present on the surface of the cherry tomatoes used, whereas the counts of total aerobic bacteria in the PAW-treated groups were below the detectable levels (<0.42 log_10_ CFU/g) until day 6. It is believed that the reactive species generated in PAW possess antimicrobial activities against a broad range of microbes, including Gram-positive and Gram-negative bacteria, viruses, and fungi [23,34,45]. Initial counts of yeasts and molds in the unwashed and DW-washed groups were approximately 0.6 and 1.8 log_10_ CFU/g, respectively, whereas the counts of yeasts and molds in the PAW-washed group remained below the detectable levels (<0.72 log_10_ CFU/g) until the final day (Figure 6). Finally, compared with those in the unwashed group, the final counts of total aerobic bacteria and yeasts and molds in the PAW-washed group reduced by 1.73 ± 0.12 and 1.5 ± 0.15 log_10_ CFU/g, respectively. These results are consistent with those of other reports that PAW-induced inactivation of fungal cells is less compared to that of bacteria cells [22,23,34]. In addition, coliform bacteria and *E. coli*, indicators of water quality and hygiene, were not present in all groups, and thus have not been shown in Figure 6. Hence, it was concluded that PAW exerts inhibitory effects on aerobic bacteria, yeasts, molds, and coliforms present on the surface of cherry tomatoes. In addition, these results indicate that when the initial microbial counts in fresh produce are low, it is difficult for these microbes to proliferate even at room temperature, regardless of washing or using disinfectants. Besides the microbial safety aspects, the effects of PAW on quality of fresh produce is also an important factor. In particular, color is considered the primary quality parameters for consumers. However, many studies have already reported that no significant change was observed in the color of fresh produces treated with PAW [15,22,34]. Thus, we have focused on the microbial inactivation effect on PAW without quantitative color measurement in this study. Therefore, washing fresh produce with PAW is essential to disinfect fresh produce containing high levels of microbial contamination for enhancing food safety from farm to fork and prolonging their shelf-life.

### 3.5. Cytotoxicity of PAW

To check whether PAW treatment of fresh produce without additional washing steps carries the risk of adversely affecting mammalian cells, thereby leading to cytotoxicity, we tested whether PAW15 treatment induces cytotoxicity in RAW264.7 cells. The viability of RAW264.7 cells treated with undiluted PAW15 was 99.3 ± 0.5%, which was not significantly different from that in the DW-washed group, indicating no cytotoxicity (Figure 7). Despite reports that exposure to reactive species may induce oxidative or nitrosative stress in mammalian cells [46], many studies have reported that H_2_O_2_ or NO_3_^−^ are not cytotoxic, even at tenfold higher concentrations than those generated in PAW [47,48,49]. Our results agreed with those of other studies. Based on the results of this cytotoxicity test, it can be inferred that eating fresh produce washed with PAW will not adversely affect human health. Moreover, there is no risk of environmental contamination even if the used wash water is released without separate disinfection treatment. In addition, compared with the effects of chlorinated water treatment, PAW treatment does not leave any residual chemicals or odors but enables the inactivation of native microbes or inoculated pathogens on the surface of cherry tomatoes. Accordingly, it can be used as a substitute for traditional chlorine-based disinfectants.

## 4. Conclusions

Herein, we show that PAW-washing treatment has the potential to inactivate microbes present on cherry tomatoes and in the used wash solution. The pH values and concentrations of reactive species (H_2_O_2_, NO_2_^−^, and NO_3_^−^) of PAW were measured after different lengths of plasma treatment times. The optimal HT following the exposure of bacteria to PAW15 was determined as 5 min. The microbial inactivation efficacy of PAW15 was compared with those of mimic solutions containing comparable pH values and reactive species concentrations. The inactivation efficacy of PAW15 was 1.3 log_10_ CFU/g higher than those of mimic solutions bearing reactive species, likely due to the synergistic effect of peroxynitrous acid, produced by the reaction of H_2_O_2_ and NO_2_^−^, and other factors inherent in PAW. Compared with the counts of inoculated foodborne pathogens in cherry tomatoes washed with sterile DW, PAW15-washing treatment achieved a significant reduction in the counts of these pathogens ranging from 1.6–2.6 log_10_ CFU/g (*p* ≤ 0.05), but none of these pathogens were detected in the used wash solution. PAW-washing treatment was also effective in inhibiting the growth (<1 log_10_ CFU/g) of bacteria, yeasts, and molds present on the surface of cherry tomatoes during storage for 6 days at 25 °C, although the treatment exhibited no toxicity in RAW264.7 cells. These results support the use of PAW as a safe, effective, and eco-friendly wash solution for microbial decontamination with a wide host range and without the risk of cross contamination in the fresh produce industry. However, to ensure the application of PAW as an alternative wash solution in the industrial washing process, it may be necessary to scale up plasma generation and optimize operation conditions of the washing process.

## Figures and Tables

**Figure 1 foods-12-02461-f001:**
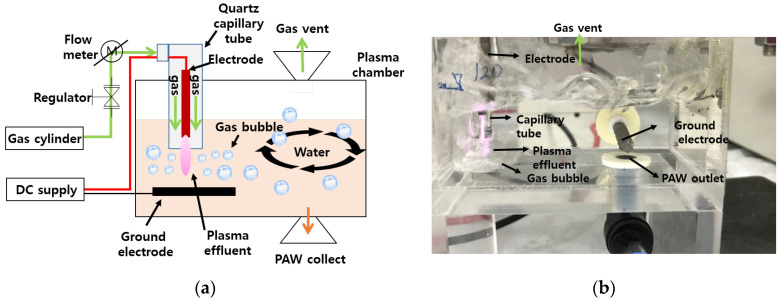
Schematics showing the configuration (**a**) and the photograph (**b**) of the plasma jet used to generate plasma-activated water.

**Figure 2 foods-12-02461-f002:**
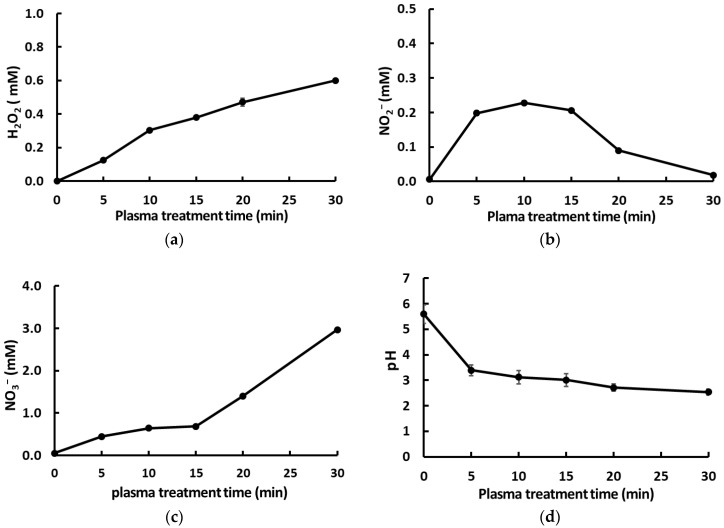
Effects of varying plasma treatment times on reactive species concentrations and the pH value of plasma-treated water. Changes in the species concentrations of (**a**) hydrogen peroxide (H_2_O_2_), (**b**) nitrite (NO_2_^−^), and (**c**) nitrate (NO_3_^−^) and in (**d**) the pH value of water in response to increasing plasma treatment times. The concentration of H_2_O_2_ shows a continuous decreasing trend, whereas those of NO_2_^−^ and NO_3_^−^ initially increase and then decrease with increasing lengths of plasma treatment times. The pH value of water shows a continuous decreasing trend with increasing plasma treatment times. The results are the means of three replications, and vertical bars indicate standard deviation.

**Figure 3 foods-12-02461-f003:**
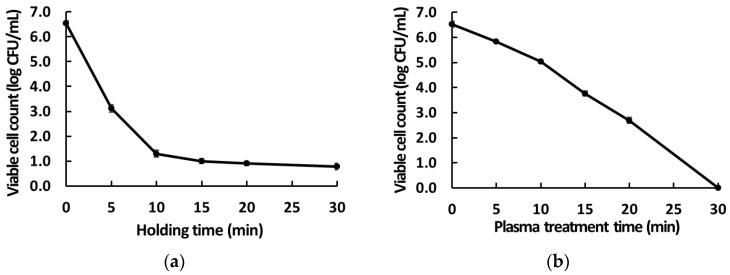
Viable cell counts of *B. cereus* after (**a**) different holding (contact) times with plasma-activated water 15 (PAW15, generated via plasma activation for 15 min) and (**b**) 5 min treatments with PAW generated via exposure to plasma discharge for different lengths of time. The results are the means of three replications, and vertical bars indicate standard deviation.

**Figure 4 foods-12-02461-f004:**
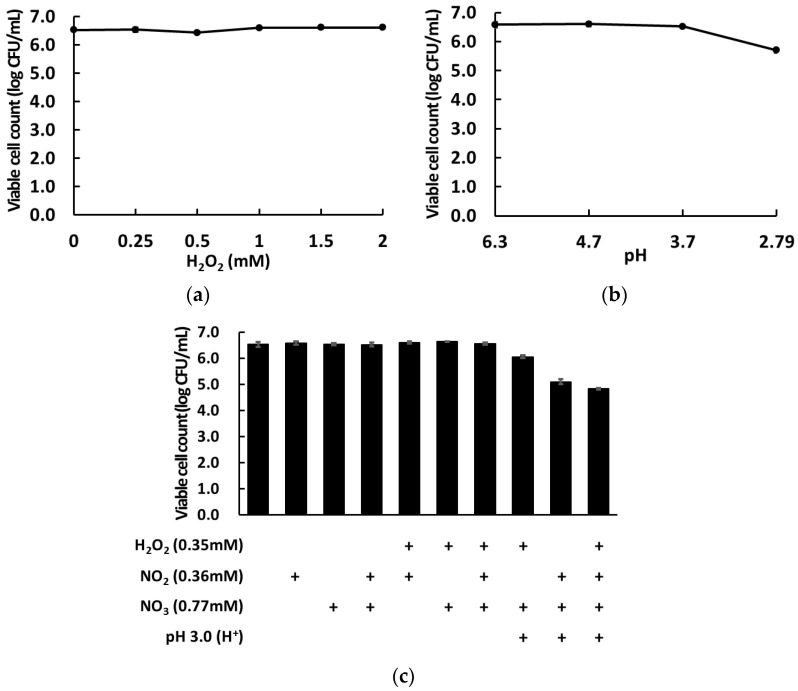
Changes in viable cell counts of *Bacillus cereus* in response to increasing (**a**) concentrations of hydrogen peroxide (H_2_O_2_), (**b**) pH, (**c**) various combinations of artificially generated reactive species in mimic solutions with concentrations of H_2_O_2_, NO_2_^−^, and NO_3_^−^ species and pH values comparable to those of PAW15. The results are the means of three replications and vertical bars indicate standard deviation.

**Figure 5 foods-12-02461-f005:**
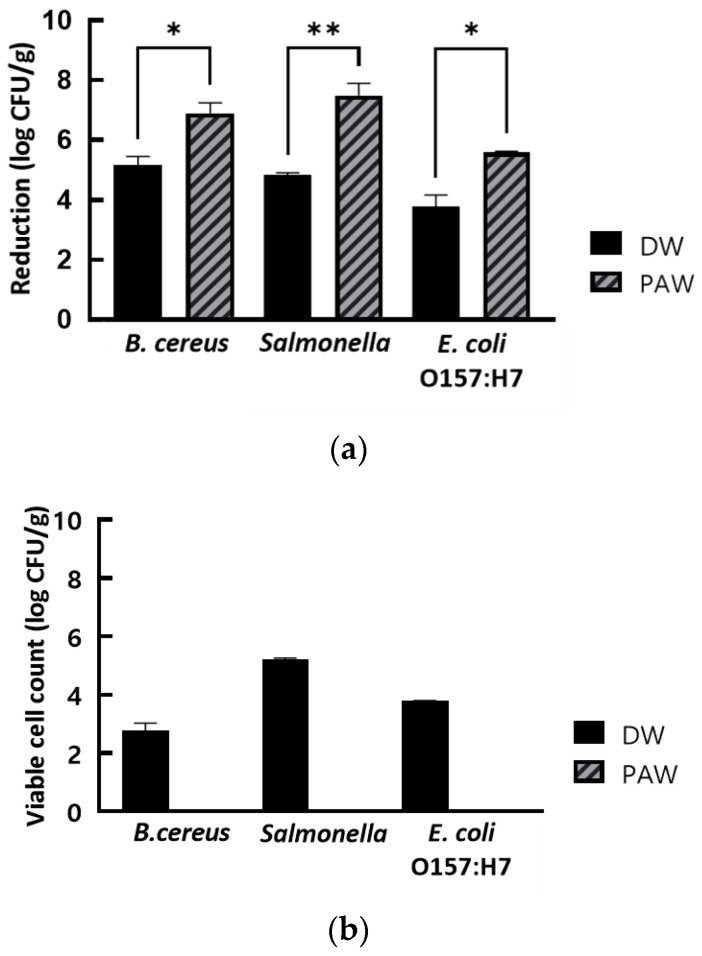
(**a**) Reduction in viable cell counts of foodborne pathogens inoculated on the surface of cherry tomatoes and (**b**) the survival of foodborne pathogens in used wash solutions obtained after washing the cherry tomatoes, subjected to plasma-activated water and sterile distilled water treatments. The results are the means of five replications and vertical bars indicate standard deviation. * *p* < 0.05, ** *p* < 0.005.

**Figure 6 foods-12-02461-f006:**
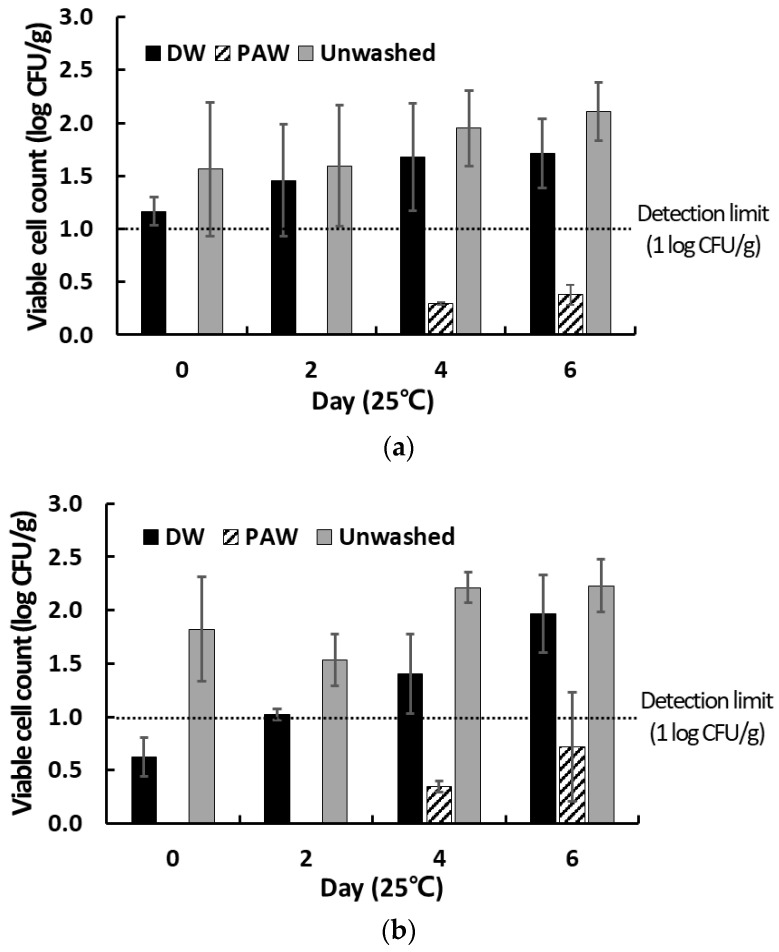
Effects of PAW-washing treatment on the counts of (**a**) total aerobic bacteria and (**b**) yeasts and molds in cherry tomatoes during storage at room temperature. The results are the means of three replications and vertical bars indicate standard deviation.

**Figure 7 foods-12-02461-f007:**
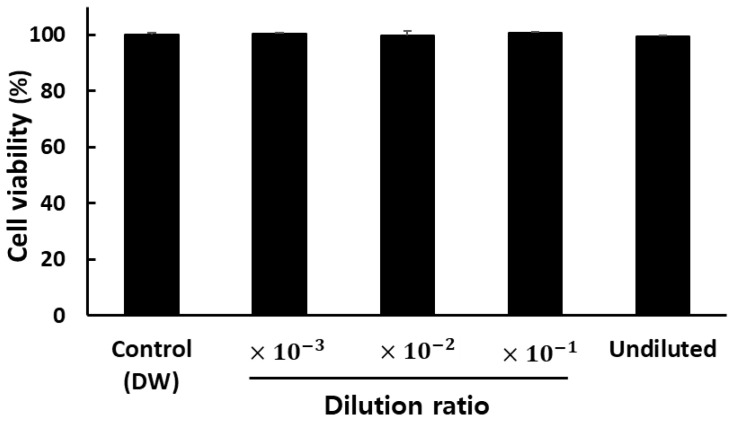
Cell viability was investigated via the MTT assay after treatment with PAW15 (generated via plasma activation for 15 min) and its dilutions and distilled water as a control in RAW264.7 cells. The results are the means of five replications and vertical bars indicate standard deviation.

## Data Availability

The data is contained within the article (or Appendix A).

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
