# Peer review of "Effects of Plasma-Activated Water Treatment on the Inactivation of Microorganisms Present on Cherry Tomatoes and in Used Wash Solution"

_foods, 2023, doi:10.3390/foods12132461_

Round 1
Reviewer 1 Report
1. Figure 1 divide into two (Figure 1.a and Figure 1.b). First, it is necessary to indicate the components in the diagram.
2. Describe the source of DC, because it needs to be clear how the plasma is generated.
3. The word (DifcoTM) from line 150 should be removed as it is written in line 136
4. The word (Sigma-Aldrich) in line 164 should also be removed since it is written in line 162.
5. If the authors had results of killing bacteria with a PAW10, it would be excellent for comparison. This, is under the results in Figure 2.
6. In line 242, the authors indicate “previous studies, i.e.,” a reference does not support this; it is convenient to place it.
No comment
Author Response
1. Figure 1 divide into two (Figure 1.a and Figure 1.b). First, it is necessary to indicate the components in the diagram.
→As the reviewer suggested, Figure 1 is divided into Figure 1a and Figure 1b, and the components are indicated in Figure 1b. (line 113)
2. Describe the source of DC, because it needs to be clear how the plasma is generated.
→The power source used DC voltage as described in chapter 2.3., and this DC source supplies 110V AC (50 ~ 60 Hz) and output 3 ~ 30 KV (0 ~ 10 mA). In this paper, the voltage used to discharge the plasma is 4 ~ 7 KV. This was revised to reflect the reviewer’s request. (line 107~110 in red)
3. The word (DifcoTM) from line 150 should be removed as it is written in line 136.
→ We have removed as suggested. (line 158 in red)
4. The word (Sigma-Aldrich) in line 164 should also be removed since it is written in line 162.
→ We have removed as suggested. (line 173 in red)
5. If the authors had results of killing bacteria with a PAW10, it would be excellent for comparison. This, is under the results in Figure 2.
→ As you point out, we also thought it would be better if we compared the results of killing bacteria with PAW10. I actually tested PAW10 because we thought it might be economical to use PAW10 and got almost the similar results as that of PAW15. However, because the bacterial killing effect of PAW10 was not stable, unfortunately we had no choice but to present only the results of PAW15 in this manuscript.
6. In line 242, the authors indicate “previous studies, i.e.,” a reference does not support this; it is convenient to place it.
→ We have added references (15, 22, 31 (new add), 32) and changed reference numbers throughout the manuscript. (line 255 in red)

Reviewer 2 Report
Effects of Plasma-Activated Water Treatment on the Inactivation of Microorganisms Present on Cherry Tomatoes and in Used Wash Solution
This manuscript compensates the data of toxicity of plasma activated water (PAW) and showed excellent results of antibacterial activity. However, there are several points authors need to improve. The main points are the bacteria used in the study and evaluation of active species of PAW.
Following is my comments:
1. Bacillus cereus is not a common pathogen associated with tomatoes. Could authors list some examples of Bacillus cereus associated with foodborne illness outbreaks of tomatoes?
2. 2.3: the volume of PAW is 120 mL, which is less than several previous studies of PAW. Therefore, it is contradictory to the description in the Introduction for industrial application.
3. 2.4: PAW also generates ozone that is a proved antibacterial substance. Why did not author check ozone?
4. 2.5: Authors determine the holding time first, then activation time. Could authors describe why following the sequence since many studies determine activation time first, then holding (treatment) time.
5. 2.7: Many short-lived substances only exist during the activation of PAW. In addition, the UV light generated by the plasma electrode in the PAW also provide antibacterial effect. Thus, placing tomatoes in PAW while the plasma electrode was activating is more effective than placing tomatoes in the collected PAW in which short-live substance and UV do not exist.
6. The abbreviation of Salmonella enterica subsp. enterica serovar Typhimurium should be S. Typhimurium. (line 169)
7. 2.8: In the section, 120 mL water was used but 2000 mL of PAW tested in this section.
8. Fig. 3b: it is better to indicate the 5-min treatment in the legend.
9. Why did authors use Bacillus cereus for the test of mimic solutions? E. coli and Salmonella spp. are more frequently associated with tomatoes. It is also better to test Gram positive and negative bacteria in the same research.
10. Line 312: H2O2
11. When using water washing as the baseline, the bacterial reductions of PAW were lower than many studies of PAW. Could authors provide reasons?
12. B. cereus, a Gram positive bacteria, showed greater reduction than Salmonella and E. coli, both Gram negative bacteria. These results were contradictory to many studies of PAW. Could authors explain these results?
13. How did authors determine the bacterial counts in the treated water? Authors did not describe the procedures.
14. Many researches used PAW to inactivate pathogens on food items. Some of them also measured concentrations HNO3-, HNO2-, and H2O2 as well as the analyses of pH, oxidative-reduction potential (ORP), UV spectrum, and electron paramagnetic resonance (EPR) to detect singlet oxygen and hydroxyl ions. Recommend authors to describe those researches.

Author Response
1. Bacillus cereus is not a common pathogen associated with tomatoes. Could authors list some examples of Bacillus cereus associated with foodborne illness outbreaks of tomatoes?
→ As the reviewer notes, B. cereus is not a common pathogen on tomatoes and few cases have been associated with food illness outbreaks in tomatoes. However, B. cereus is widely distributed in environments such as soil and plants, and is frequently isolated from tomatoes even though the contamination level was less than 1 log CFU/g. So it was used as target bacteria in this study as potential risk bacteria. In addition, the references below are presented as examples of tomato-related B. cereus outbreaks. (line 89-93 in red)
https://ucfoodsafety.ucdavis.edu/sites/g/files/dgvnsk7366/files/inline-files/223995.pdf
2. 2.3: the volume of PAW is 120 mL, which is less than several previous studies of PAW. Therefore, it is contradictory to the description in the Introduction for industrial application.
→The basic configuration of the plasma we used in this manuscript is the device that manufactures PAW with a 120 mL DW as described in section 2.3. As the reviewer pointed out, we agree that the current manufacturing volume is insufficient for industrial application of PAW. However, this manuscript is a preliminary investigation into the effects on microbes and fresh produce when 120 mL water is discharged to generate PAW by plasma. In this manuscript, we conducted an experiment applied to cherry tomatoes using 2,000 mL PAW. And 2,000 mL volume PAW was fabricated in parallel using 10 units (or devices) of this device to simultaneously manufacture PAW, and the unit configuration device was the same as in section 2.3. And new large scale plasma device (4,000 mL volume water tank) for industrial application is being developed by the company (Plasma Holdings Co.Ltd., Korea) that supplied the devices for this study. (line 110~111 in red)
3. 2.4: PAW also generates ozone that is a proved antibacterial substance. Why did not author check ozone?
→That’s good point. As first, we also knew empirically that a significant amount of ozone gas was generated from the plasm device. However, when dissolved ozone in PAW was measured, it did not appear to be undetectable. We guess that this is due to the reaction with already formed hydrogen peroxide (Peroxon process), so ozone disappeared within seconds, showing an unmeasurable amount of ozone (ref: Plasma Chemstry and Catalysis in Gases and Liquids, Chapter 7.2.2.2, Petr Lukes, et.al., Wiley-VCH Verlag GmbH & Co, KGaA, 2012). Thus, we though that the PAW’s effect on microorganisms would be an active species in exclude the dissolved ozone that could exist for several minutes after the plasma discharge was terminated, and this active species was guessed to be hydrogen peroxide and nitrous acid in an acidic environment.
4. 2.5: Authors determine the holding time first, then activation time. Could authors describe why following the sequence since many studies determine activation time first, then holding (treatment) time.
→The Korea Ministry of Food and Drug Safety recommends immersion within 5 minutes when washing fresh produces with disinfectant, so we considered that it should be determined first because holding time can affect freshness in fresh produce industry applications.
5. 2.7: Many short-lived substances only exist during the activation of PAW. In addition, the UV light generated by the plasma electrode in the PAW also provide antibacterial effect. Thus, placing tomatoes in PAW while the plasma electrode was activating is more effective than placing tomatoes in the collected PAW in which short-live substance and UV do not exist.
→ Instead of PAW, the direct method of treating with plasma discharge is well known, and its antibacterial effect is also better than PAW, as the reviewer pointed out. However, the chamber must be made in consideration of food volume, and the number and size of electrodes. So, at this point, we first tried the in-direct method using PAW to process a lot of food at the same time. When the plasma discharge was terminated, the most active species disappeared, so it was thought that the antimicrobial activity would be lowered. However, PAW, the water generated after the plasma discharge, also had antibacterial activity.
6. The abbreviation of Salmonella enterica subsp. enterica serovar Typhimurium should be S. Typhimurium. (line 169)
→ We have corrected as suggested. (line 178 in red)
7. 8: In the section, 120 mL water was used but 2000 mL of PAW tested in this section.
→ As the reviewer suggested, we added a method for preparing 2000 mL in section 2.3. (line 110~111 in red)
8. Fig. 3b: it is better to indicate the 5-min treatment in the legend.
→ We have corrected as suggested. (line 278 in red)
9. Why did authors use Bacillus cereus for the test of mimic solutions? E. coli and Salmonella spp. are more frequently associated with tomatoes. It is also better to test Gram positive and negative bacteria in the same research.
→ As the holding time and plasma treatment time for PAW washing were previously determined with B. cereus, the microbial inactivation effect of mimic solution had to be tested with the same strain in order to match the experimental results. As the reviewer comments, it would be nice if experiments on Gram negative could be added, but unfortunately we do not have results on them.
10. Line 312: H2O2
→ We have corrected it. (line 331)
11. When using water washing as the baseline, the bacterial reductions of PAW were lower than many studies of PAW. Could authors provide reasons?
→ As pointed out by the reviewer, the bacterial reductions of PAW (higher approximately 1.6~2.7 log CFU/g than reductions of water treatment) in our result were lower than many studies of PAW. We think the reason is because the surface roughness of the fresh produces compared is different. It has already been reported that bacteria on a soft surface like cherry tomatoes can be easily removed with just water, whereas it is difficult to remove bacteria on a rough surface like strawberries. (reference [13])
12. B. cereus, a Gram positive bacteria, showed greater reduction than Salmonella and E. coli, both Gram negative bacteria. These results were contradictory to many studies of PAW. Could authors explain these results?
→ We also expected that PAW under the conditions in which B. cereus, a Gram positive bacteria is inactivated would show greater reduction when applied to Gram negative bacteria. However, as in this manuscript, it was different from what was expected. We repeated this experiment more than 10 times and got the same results. We think that the biggest reason could be the difference in strains. If we used strains other than B. cereus ATCC 14579, Salmonella Typhimurium ATCC 43971, and E. coli O157:H7 NCCP 1109, we might have similar results to other studies. So, we are going to apply various strains of the same species in the ongoing scale-up experiment.
13. How did authors determine the bacterial counts in the treated water? Authors did not describe the procedures.
→ We have written a description of the procedures on line 174~175 in red.
14. Many researches used PAW to inactivate pathogens on food items. Some of them also measured concentrations HNO3-, HNO2-, and H2O2 as well as the analyses of pH, oxidative-reduction potential (ORP), UV spectrum, and electron paramagnetic resonance (EPR) to detect singlet oxygen and hydroxyl ions. Recommend authors to describe those researches.
→ That’s good point. We also measured OPR, pH, conductivity and attached the data just below. This result was similar to that of many previous studies. This is a well-known result, and it is omitted because it is judged that it is not helpful in explaining the results of this paper. In addition, UV spectrum and EPR are useful for confirming the existence of short-lived species in direct method, and were not measured because they did not help to confirm the long-loved species in this paper.

Round 2
Reviewer 2 Report
This manuscript has improved. However, authors did not response one point correctly.
14. For the following comment, what I suggest was to expand the discussion including the previous studies reporting HNO3-, HNO2-, and H2O2 as well as the analyses of pH, oxidative-reduction potential (ORP), UV spectrum, and electron paramagnetic resonance (EPR). That should give readers more complete understanding about the mechanism of PAW.
14. Many researches used PAW to inactivate pathogens on food items. Some of them also measured concentrations HNO3-, HNO2-, and H2O2 as well as the analyses of pH, oxidative-reduction potential (ORP), UV spectrum, and electron paramagnetic resonance (EPR) to detect singlet oxygen and hydroxyl ions. Recommend authors to describe those researches.---

Author Response
14. For the following comment, what I suggest was to expand the discussion including the previous studies reporting HNO3-, HNO2-, and H2O2 as well as the analyses of pH, oxidative-reduction potential (ORP), UV spectrum, and electron paramagnetic resonance (EPR). That should give readers more complete understanding about the mechanism of PAW.
→ As the reviewer suggested, ORP, pH and conductivity results were added to the supplementary materials (line 502~504), and a discussion of these results was described in the text (line 135~137; line 234~244). We have added references ([19, 31, 32]) related to this and changed reference numbers throughout the manuscript. In addition, UV spectrum and EPR are useful for confirming the existence of short-lived species in direct method, and were not measured because they did not help to confirm the long-loved species in this paper.
